# Machine Learning-Based Early Warning Level Prediction for Cyanobacterial Blooms Using Environmental Variable Selection and Data Resampling

**DOI:** 10.3390/toxics11120955

**Published:** 2023-11-23

**Authors:** Jin Hwi Kim, Hankyu Lee, Seohyun Byeon, Jae-Ki Shin, Dong Hoon Lee, Jiyi Jang, Kangmin Chon, Yongeun Park

**Affiliations:** 1School of Civil and Environmental Engineering, Konkuk University, Gwangjin-gu, Seoul 05029, Republic of Korea; jinhwi25@naver.com (J.H.K.); haeckel@konkuk.ac.kr (H.L.); shbyeon1@gmail.com (S.B.); 2Busan Region Branch Office of the Nakdong River, Korea Water Resources Corporation (K-Water), Saha-Gu, Busan 49300, Republic of Korea; shinjaeki@gmail.com; 3Department of Civil and Environmental Engineering, Dongguk University-Seoul, 30, Pildong-ro 1-gil, Jung-gu, Seoul 04620, Republic of Korea; leedonghoon@dongguk.edu; 4Division of Atmospheric Sciences, Korea Polar Research Institute, 26, Songdomirae-ro, Yeonsu-gu, Incheon 21990, Republic of Korea; j.jiyi19@kopri.re.kr; 5Department of Environmental Engineering, Kangwon National University, Gangwon-do, Chuncheon 24341, Republic of Korea; kmchon@kangwon.ac.kr; 6Department of Integrated Energy and Infra System, Kangwon National University, Gangwon-do, Chuncheon 24341, Republic of Korea

**Keywords:** harmful algal blooms, alert level, feature selection, data resampling, machine learning, early warning

## Abstract

Many countries have attempted to mitigate and manage issues related to harmful algal blooms (HABs) by monitoring and predicting their occurrence. The infrequency and duration of HABs occurrence pose the challenge of data imbalance when constructing machine learning models for their prediction. Furthermore, the appropriate selection of input variables is a significant issue because of the complexities between the input and output variables. Therefore, the objective of this study was to improve the predictive performance of HABs using feature selection and data resampling. Data resampling was used to address the imbalance in the minority class data. Two machine learning models were constructed to predict algal alert levels using 10 years of meteorological, hydrodynamic, and water quality data. The improvement in model accuracy due to changes in resampling methods was more noticeable than the improvement in model accuracy due to changes in feature selection methods. Models constructed using combinations of original and synthetic data across all resampling methods demonstrated higher prediction performance for the caution level (L-1) and warning level (L-2) than models constructed using the original data. In particular, the optimal artificial neural network and random forest models constructed using combinations of original and synthetic data showed significantly improved prediction accuracy for L-1 and L-2, representing the transition from normal to bloom formation states in the training and testing steps. The test results of the optimal RF model using the original data indicated prediction accuracies of 98.8% for L0, 50.0% for L1, and 50.0% for L2. In contrast, the optimal random forest model using the Synthetic Minority Oversampling Technique–Edited Nearest Neighbor (ENN) sampling method achieved accuracies of 85.0% for L0, 85.7% for L1, and 100% for L2. Therefore, applying synthetic data can address the imbalance in the observed data and improve the detection performance of machine learning models. Reliable predictions using improved models can support the design of management practices to mitigate HABs in reservoirs and ultimately ensure safe and clean water resources.

## 1. Introduction

Toxic harmful algal blooms (HABs) cause various environmental problems in aquatic ecosystems, including public health threats, massive fish deaths, drinking water safety problems, increased wildlife mortality, and the destruction of aquatic habitats [1,2]. The recent rise in water temperature owing to climate change and the increase in nutrient discharge caused by human activity have promoted the growth of HABs in aquatic ecosystems [3,4,5]. In 2007, excessive algal blooms in Lake Taihu, China, affected the supply of drinking water for approximately two million people in nearby cities [6]. Furthermore, European countries, such as France, the United Kingdom, and Italy, suffer from social, economic, and environmental problems caused by HABs in coastal and inland areas [7,8,9]. These events suggest that toxic HABs can threaten public health and regional economies by contaminating drinking water, fish, and shellfish.

Therefore, the excessive growth of HABs across all regions is a significant global concern related to water quality management [10]. Therefore, many countries around the world, including South Korea, have conducted research and introduced policies and activities to solve the algal bloom problem to protect aquatic ecosystems, reduce public health threats, and secure safer water resources. As part of this, algal alert warning systems have been introduced and used in many countries to respond quickly to high-level algal blooms with HABs [11,12,13]. Recently, the Food and Agriculture Organization of the United Nations (FAO), in collaboration with the Intergovernmental Oceanographic Commission (IOC) and the International Atomic Energy Agency (IAEA), developed technical guidelines for implementing early warning systems for HABs that affect food safety or security [14]. Furthermore, algal alert warning systems serve as an important indicator for monitoring and managing algal blooms in terms of water quality management. They provide monitoring and management sequences to government officials, drinking water treatment plant operators, and water quality managers to help them make decisions [15].

In South Korea, a large-scale national project was implemented to dredge rivers and install eco-friendly weirs to increase the water storage capacity and restore the ecosystems of the country’s four major rivers. However, since 2012, the flow velocity of rivers between weirs has decreased, leading to an increase in the frequency and intensity of algal blooms with HABs and risk of drinking water pollution [16,17]. In South Korea, an algal alert warning system is currently in place at 29 stations along four major rivers and reservoirs. The algal alert level of the system is determined based on the concentration of harmful algal cells. Therefore, the system focuses on the postblooming response rather than predicting the algal alert level. If the algal alert level can be predicted, it would be possible to respond before the occurrence of HABs with proactive water quality management.

In recent years, various studies have been conducted on data-driven models, which are easier to construct than numerical models [18]. However, the frequency of water quality monitoring for HABs is typically weekly or monthly [19,20], which makes it challenging to acquire sufficient data to train machine learning models. In addition, the occurrence of algal blooms typically has a seasonal pattern; algal blooms rarely occur in cold winters when the temperature is low and usually occur from spring to autumn when the temperature rises [21]. The magnitude of algal blooms has an uneven distribution and is characterized by sporadic occurrences [11]. For this reason, the distribution of data is imbalanced when classified based on the concentration or alert level of harmful algae. Shin et al. [22] collected and analyzed the distribution of algal alert levels at 13 monitoring stations in a reservoir and reported that the distribution was imbalanced at nine of the stations. Training machine learning models using imbalanced data can lead to accurate predictions at the majority alert level and inaccurate predictions at the minority level. However, accurate predictions for algal blooms with high concentrations that occur infrequently can be utilized as more important information than predictions for low concentration blooms in terms of water quality [23]. Furthermore, the results of supervised machine learning models are dominated by the quality and quantity of the data used in the training step. Therefore, class imbalance data in classification models reduce the ability to predict minority classes, and basic machine learning algorithms designed to improve overall prediction performance more accurately predict instances of the majority class than the minority class [24,25].

Recently, several studies have been conducted to solve the problem of data imbalance in statistical models. Choi et al. [26] solved the data imbalance problem using the synthetic minority oversampling technique (SMOTE), an oversampling method, and predicted the chlorophyll-a concentration in the Daechung reservoir in South Korea using a convolutional neural network model. Jeong et al. [27] considered SMOTE to solve data imbalance and predicted cyanobacterial cell density in eight water supply reservoirs in South Korea using machine learning models such as random forest (RF) and extreme Gradient Boosting. Bourel et al. [28] considered three under- and oversampling methods, including SMOTE, and predicted fecal coliforms on 21 beaches in Uruguay using various machine learning models. Despite the existence of such studies, studies specifically addressing imbalanced data related to harmful algal blooms, chlorophyll-a, nutrients, and specific environmental problems in the field of aquatic ecosystems are limited. In addition, few comprehensive studies have simultaneously addressed imbalanced data related to harmful algal blooms and feature selection for input variables.

Therefore, the impact of feature selection and data imbalance on machine learning models must be evaluated. The specific objectives of this study were to (1) acquire environmental variables, including water quality, hydrologic, and meteorological data, as input variables and apply feature selection methods to identify appropriate environmental variables, (2) solve data imbalance by generating synthetic data for minority classes using various resampling methods based on measurement data, (3) develop the algal alert warning system that can predict the algal alert levels in advance using artificial neural network (ANN) and RF models, and (4) evaluate differences in feature selection and resampling methods for improving prediction accuracy in minority classes.

## 2. Materials and Methods

### 2.1. Site Description

The Geum River is a major river in South Korea with agricultural and industrial functions. The shape and flow system of the river has changed since the construction of a multifunctional weir in 2012 [11], increasing the retention time of the flow rate [29]. As a result, blooms have expanded to the middle and upper reaches of rivers [30]. In addition, algal blooms, including HABs, have been continuously reported in the BJR [31]. The study area was the Baekje reservoir (BJR), located at the mid-stream of Geum River between 126°56′20″ E and 127°05′55″ E longitude and 36°19′07″ N and 36°27′45″ N latitude. The river width between the BJR and Gongju Reservoir (GJR) is 290–570 m, which is relatively large in South Korea (Figure 1). The BJR weir, which is a prediction point for algal alert levels, is located downstream of the study area and the GJR, which collects the cell density of cyanobacteria as an input variable, is located upstream of the study area. The main land-use type in the environs of the BJR is agricultural.

### 2.2. Data Acquisition

Seven water quality variables such as cyanobacteria cell density, total dissolved nitrogen concentration (TDN), nitrate concentration (NO_3_-N), ammonium concentration (NH_4_-N), total dissolved phosphorus concentration (TDP), phosphate concentration (PO_4_-P), and conductivity (Cond) in the BJR were collected by the Korea Ministry of Environment from a monitoring station which was located 500 m upstream of the weir (Table 1). The average monitoring interval was 8 days and ranged from 4 to 62 days owing to irregular sampling caused by weather conditions, sampling management officers, and reservoir conditions. The algal alert levels of the BJR as an output variable were classified into three levels according to the classification criteria of the algal alert warning system implemented in South Korea based on cyanobacteria cell density [32]: normal level (<1000 cells/mL, L-0), caution level (≥1000 cells/mL and <10,000 cells/mL, L-1), warning level (≥10,000 cells/mL and <1,000,000 cells/mL, L-2), and blooming level (≥1,000,000 cells/mL, L-3). Four hydrological and three meteorological variables were monitored by the Korea Water Resource Corporation and the Korea Meteorological Administration. Daily hydrological and meteorological data, including air temperature, wind speed, water level, total inflow, total discharge, and total hydropower plant discharge, were used as average values between water quality monitoring events, and precipitation was used as the cumulative precipitation. In addition, cyanobacterial cell density measured at the GJR water quality monitoring station upstream of the BJR was used as an input variable to consider the connectivity between the two reservoirs for predicting algal alert levels. In this study, a total of 429 datasets were collected over 9 years from 2013 to 2021 (Figure 2A), but 345 datasets were chosen to develop a machine learning model. Datasets were excluded from the winter season (January, February, and December) in South Korea because of the impossibility of monitoring frozen rivers and the lack of cyanobacterial growth at low temperatures. During this period, the algal alert levels corresponding to L-1 and L-2 were zero. Therefore, we developed a machine learning model using 345 monitoring data points from March to November. All data were collected over 9 years (2013–2021) (Figure 2A). Table 1 lists the 14 environmental variables considered as input variables.

### 2.3. Feature Selection for Algal Alert Levels

We statistically identified the relationship between cyanobacteria cell density, which determines algal alert levels, and input variables, including water quality, hydrodynamics, and meteorological variables, using linear and non-linear variable selection methods (Figure 2B). For the linear variable selection method, a simple linear regression analysis was used to analyze the input and output variables one-to-one. Simple linear regression was used to statistically test the dependence between variables [33] and the dependent input variables were selected based on statistical significance (*p* < 0.05). For the non-linear variable selection method, mutual information (MI) was used to measure the degree of relatedness between the output and input variables. MI is interpreted as the amount of information shared between variables, regardless of the average value and variance, and is based on information theory on a methodologically established basis [34]. The larger the MI, the higher the dependence on the probability distribution between variables. At an MI of 0, the relationship between variables is independent. Table 1 shows the statistical significance and MI scores for each of the 14 input variables.

### 2.4. Resampling Methods for Imbalanced Datasets

In data-driven models, including machine learning, deep learning, and linear statistical models, the imbalanced distribution of the output variable to be predicted results in the biased learning of the model because the accuracy is dominated by the amount and quality of the original dataset [35]. A total of 345 cyanobacteria cell density data collected from the BJR with algal alert level criteria were classified into L-0 (269; 78.0%), L-1 (47; 13.6%), and L-2 (29; 8.4%), respectively. The distribution of algal alert levels was sufficiently unbalanced to affect the model training. In a previous study [36], we used adaptive synthetic sampling (ADASYN) to generate synthetic data for algal alert levels corresponding to L-1 and L-2 based on observational data, addressing the imbalance of the data and improving the accuracy of the machine learning model. In the previous study, the amount of data was increased using synthetic data to resolve the data imbalance. In the present study, the amount of data increased and a method of reducing the majority class to the minority class was considered.

Oversampling involves creating copies of existing samples or adding more samples with values similar to those of a minority class [37]. However, oversampling can increase the size of the training dataset, resulting in additional computation time and potential overfitting of the model [38]. Undersampling involves the removal of samples from the majority class until a balance is achieved between the minority and majority classes. Therefore, during the training step, the reduced amount of data can improve the computation time for weight calculation and address storage-related issues, making the overall model implementation more efficient, which may improve the predictive accuracy of the model [39]. However, using undersampling, it may be challenging to improve the imbalance in predicting algal alert levels for relatively small datasets, such as that used in this study. To address these issues, hybrid sampling methods, such as ENN, which combine oversampling and undersampling have been proposed [40]. The resampling methods used were as follows: (1) random oversampling (ROS), SMOTE, and ADASYN as oversampling methods; (2) cluster centroid undersampling (CC) and random undersampling (RUS) as undersampling methods; and (3) synthetic minority oversampling technique–edited nearest neighbor (ENN) and synthetic minority oversampling technique–Tomek link (Tomek) as hybrid sampling methods. The detailed resampling methods are described in Appendix A.

### 2.5. Construction of Machine Learning Models and Evaluation of Model Accuracy

Figure 2 shows a flowchart of the study process in the order of data preparation, synthetic data generation and application, two machine learning model constructions, and model comparison. During the data acquisition and preprocessing stages, data on algal alert levels were collected as output variables, and water quality and hydrodynamic and meteorological data were collected as potential environmental variables affecting algal blooming (Figure 2A). We determined the input variables using the linear and non-linear variable selection method between each input variable and output (Figure 2B). We modified the selected input and output variables to focus on predicting future algal alert levels (Figure 2C). In other words, the measured value of the output variable for a specific algal alert level was matched with the values of previously measured input variables in the monitoring conducted at an average interval of eight days. For example, the algal alert level measured on 23 April 2013, was considered the output variable of the input variables measured on 15 April 2013. These variables comprised a single dataset. In this preprocessing, the prediction of future algal alert levels using the current input variables was reflected in the training steps of the two machine learning models.

In the dataset reconstruction stage, all datasets were randomly extracted into training (70%) and test (30%) datasets (Figure 2D). For each resampling method, synthetic data generated based on the training dataset were added to the dataset used in the training step (Figure 2E). The dataset at the test step for all of the prediction models was used as the original dataset without adding synthetic data. Therefore, a total of 24 cases, each possessing different sets of data, were generated considering variable selection methods and resampling methods for training and testing of the models: (1) eight cases consisted of the original dataset without variable selection and seven datasets with generated synthetic data based on the original data for each resampling method, (2) eight consisted of the original dataset with the linear variable selection method and seven datasets with seven resampling methods, and (3) eight consisted of the original dataset with the non-linear variable selection method and seven datasets with seven resampling methods (Figure 2F).

In the construction and evaluation stages of the algal alert level prediction model, the datasets, excluding the respective test datasets from the 24 generated cases, were randomly extracted into training (75%) and validation (25%) datasets. The training datasets were used to train the model and optimize the hyperparameters (Figure 2G). Test datasets were used to evaluate the performance of each model constructed using the resampling methods. In this study, two prominent machine learning models, ANN and RF, were utilized to predict the alert levels for harmful algal blooms. Machine learning models, known for their powerful computational techniques, are useful for predicting specific phenomena and interpreting complex relationships in the environment [41,42]. In addition, ANN and RF models are representative machine learning models which assess the impact of imbalanced data on predictive performance, making it more convenient for other researchers to utilize the approach presented in this study. ANN and RF were optimized based on the hyperparameters of each model (Figure 2G). For ANN hyperparameters, such as the number of hidden neurons and the activation function in the hidden layer, the number of hidden neurons was optimized using a pattern search algorithm and the activation functions in the hidden layer were experimentally optimized. The activation function in the output layer was ‘softmax.’ For RF hyperparameters, such as the ensemble aggregation method, the number of ensemble learning cycles, learning rate for shrinkage, minimum leaf size, the maximum number of decision splits, and the number of predictors to select at random for each split, a random search optimization algorithm was used to optimize these hyperparameters. The ANN model structure is described in the Appendix A in our previous study [36], whereas the structure of the RF model is described in Appendix A.

Finally, the classification performances of the two models on each dataset were compared using a confusion matrix (Figure 2H). The confusion matrix is described in the Appendix A in our previous study [36]. We selected the optimized model from 100 repeated executions for each model using variable selection and resampling methods. We calculated the average accuracy of the models for each method to evaluate the overall classification performance of the two machine learning models. All processes, including statistical analysis, machine learning model configuration, and model optimization, were performed in a MATLAB (MathWorks Inc., Natick, MA, USA) environment.

## 3. Results and Discussion

### 3.1. Descriptive Analysis of Cyanobacteria and Nutrients in the BJR

Table 2 shows the results of monthly descriptive analysis for weekly cyanobacteria cell density, Chl-a concentration, and nutrient concentration in the BJR from March to November. Out of 345 events issued by the early warning system, caution (43 events) and warning (29) levels were mostly announced between July and October. The formation of algal blooms in reservoirs in East Asia, including South Korea, with monsoon climate characteristics, occurs most actively in summer [11], and these climate characteristics were reflected in the BJR. As a result of calculating the N:P ratio to identify nutrients that affect the algal growth in the BJR, the range and average value for the entire period were 5.26–240.79 and 42.5, respectively (Table 2). The N:P ratios in about 85% of samples were higher than 17, which, according to Forsberg and Ryding [43], means that primary productivity in the BJR is limited by phosphorus. Nitrogen and phosphorus are essential and influential in regulating the structure, function, and processes of ecosystems [44]. However, imbalances in the N:P ratio resulting from excessive nutrient inputs can exacerbate eutrophication in reservoirs, altering ecological structure and function and deteriorating aquatic ecosystems [45]. Therefore, the management and control of phosphorus loadings into the BJR can help suppress the occurrence of harmful algal blooms. Chl-a and phosphate concentrations from July to October, which were predominantly associated with algal bloom events corresponding to the caution and warning levels, ranged from 5.3–177.7 (an average of 50.5 µg/L) and 1–153 (an average of 31.9 µg/L), respectively. Based on Carlson [46], the nutritional status of the BJR from July to October was classified as eutrophic (Appendix A). A detailed description of the N:P ratio, Chl-a, and phosphate concentrations is given in Appendix A.

### 3.2. Selection of Input Variables and Generation of Synthetic Data

Table 1 shows the *p*-values and MI results for the 14 input variables according to the variable selection method. In the case of the dependence test, 11 variables, excluding average water level of the BJR (Wlevel), accumulated precipitation (Precip), and average wind speed (Wspeed), had a statistically significant linear dependence (*p* < 0.05) on cyanobacteria cell density; total dissolved nitrogen concentration (TDN), nitrate concentration (NO_3_-N), ammonium concentration (NH_4_-N), and conductivity (Cond) were negatively correlated, and total dissolved phosphorus concentration (TDP), phosphate concentration (PO_4_-P), average inflow rate of the BJR (Inflow), average total discharge rate of the BJR (Discharge), average discharge rate by the hydropower plant of the BJR (Dhydro), average air temperature (Atemp), and cyanobacteria cell density in the GJR (GJ-cell) were positively correlated (Appendix A). The dependence test results for phosphorus as a limiting factor for eutrophication in the BJR showed that phosphorus-related variables were positively correlated with cyanobacterial cell density, whereas nitrogen-related variables were negatively correlated. This implies that nitrogen is more abundant in the BJR than phosphorus and that a favorable N:P ratio for harmful algal blooms is formed by the inflow of phosphorus or a decrease in nitrogen in the water body. For the MI score, 12 variables were selected as input variables with statistical correlation considering nonlinearity for cyanobacteria cell density; TDN, NO_3_-N, NH_4_-N, TDP, Cond, GJ-cell, Wlevel, Inflow, Discharge, Dhydro, Atemp, and Precip had MI scores above 0 and PO_4_-P and Wspeed had scores of 0. In both variable selection methods, Wspeed, without a statistical correlation, was excluded from the input variables for predicting algal alert levels. Wong et al. [47,48] reported that wind speed affects the growth, transport, and diffusion of algal blooms. However, these studies were conducted in oceans over a wider area than the present study. Zhang et al. [49] reported that the annual average wind speed has a statistically significant correlation with the occurrence of algal blooms via regression analysis using 25 years of long-term observational data from Lake Taihu (2338 km^2^) in China. However, the yearly average wind speed was higher than that of this study area and the regression coefficient was low (−0.023~−0.027).

Considering these results, it is necessary to evaluate whether wind speed should be included as an input variable when constructing statistical models for small-scale reservoirs with characteristics similar to those in the study area. Various variable selection methods based on linear and non-linear methods can determine appropriate input variables, and the selected input variables can assist in constructing statistical models with high prediction accuracy [50]. Finally, from a total of 14 water quality, meteorological, and hydrological variables, 11 variables for the linear method and 12 variables for the non-linear method were selected as input variables to predict algal alert levels using machine learning models.

### 3.3. Improvement in Data Imbalance Using Synthetic Data

The distribution of the monitored algal alert levels used in the training step of the model from the original data was 189 (77.8%) for L-0, 33 (13.6%) for L-1, and 21 (8.6%) for L-2 (Figure 3). The overall distribution of algal alert levels was imbalanced and skewed toward the L-0. Traditionally, classification algorithms in machine learning have been used to increase the overall accuracy of the classifiers. While maximizing the overall accuracy, the model tended to focus on the majority class because of its higher weight in the distribution of the entire class [51]. For this reason, classification models can achieve high accuracy for the majority class or entire dataset, whereas they can predict poorly for minority classes. Therefore, when a dataset is imbalanced, maximizing the overall accuracy without considering the accuracy of the minority classes may not be optimal. We applied seven resampling methods of different types to improve the data imbalance: oversampling methods—ROS, SMOTE, and ADASYN; undersampling methods—CC and RUS; and hybrid sampling methods—ENN and Tomek.

Figure 3 shows the distribution of the datasets obtained using each resampling method. The datasets newly constructed using ROS, SMOTE, ADASYN, and Tomek achieved a balance between the majority class (L-0) and its data, whereas the datasets constructed using CC and RUS balanced the minority class (L-2) with the fewest samples. For ENN, oversampling was performed for L-1 and L-2 to match the data with the majority class and undersampling was performed for all classes, resulting in balanced data with 114 for L-0, 114 for L-1, and 107 for L-2. Therefore, all of the new datasets, excluding the original, were generated using a balanced distribution of algal alert levels. In the case of rare occurrence problems, identifying the minority class is often more significant than identifying the majority class and an imbalance in the dataset can lead to the generation of misleading information regarding the minority class in classification algorithms [52]. Problems such as harmful algal blooms, droughts, floods, and chemical accidents in the environment typically have a low occurrence frequency but a significant socioeconomic impact. Therefore, when analyzing these problems, it is necessary to adequately consider minority classes.

### 3.4. Comparison of Model Performance According to the Feature Selection and Resampling Methods

To assess the impact of the feature selection method on the prediction of algal alert levels, the predictive performances of the original data, original data with a linear approach, and original data with a non-linear approach were compared. Table 3 presents the performances of the ANN and RF models obtained via 100 iterations using three different datasets: the original dataset considering 14 input variables, the original dataset considering 11 input variables extracted from dependency tests, and the original dataset considering 12 input variables derived from MI scores. The key results showed that there was no clear distinction in predictive performance among the models, regardless of whether feature selection methods were applied. A detailed comparison of their performance values is provided in Appendix A. 

The ANN and RF models, using data generated by different resampling methods, were compared to evaluate the impact of data imbalance. All data used in the comparison were subjected to resampling methods without applying feature selection methods. Figure 4 shows the overall performance of each model, which was performed 100 times independently. In the training step, the overall accuracies of ANN and RF on the original dataset were relatively high. However, the overall recall for each algal alert level was unbalanced. From these results, it can be observed that the predictions for each algal alert level were unbalanced, and the accuracy was primarily influenced by L-0, indicating that it had a dominant impact on the overall performance. Furthermore, imbalanced predictions between classes in models that utilize imbalanced data can diminish the statistical reliability of the overall model accuracy [53]. Therefore, to evaluate classifiers for imbalanced data, it is essential to appropriately reflect the predictive ability of minority classes [54]. In the ANN and RF models, the predictive performance for L-1 and L-2 in the models with applied resampling methods in the training and test steps exhibited improvements compared with models utilizing the original dataset. Figure 5 shows the results of the optimal model among the models that were iteratively performed 100 times. For the ANN model, the accuracy for L-1 and L-2 improved in the training step; however, in the test step, the accuracy for L-1 improved significantly, whereas that for L-2 improved but not significantly. In the RF model, there was an enhancement in the accuracy of L-1 and L-2, with a notable improvement in accuracy, particularly for L-2. A detailed comparison of their performance values is provided in Appendix A.

### 3.5. Comparison of Model Performance According to Both Feature Selection and Resampling Methods

A total of 28 case datasets were applied to the ANN and RF models to evaluate the combined effects of the feature selection and resampling methods. Appendix A show the overall performances of the ANN and RF models, respectively, which were iteratively performed 100 times for each data type.

The overall accuracy of the two machine learning models in the training step was similar to that of the models using the original data, and the accuracy for each algal alert level was improved compared to the models using the original data. The overall recall for the model using original data was, in ANN, 66.6% for L-1 and 82.8% for L-2 and, in RF, 67.9% for L-1 and 80.6% for L-2. The range of variation in recall for L-1 and L-2 based on feature selection methods was, in ANN, 3.9–6.8% for L-1 and 3.5–4.3% for L-2 and, in RF, 0.4–4.2% for L-1 and 0.1–2.5% for L-2. The range of variation in recall based on resampling methods was, in ANN, on average, 23.3–27.4% for L-1 and 14.5–17.4% for L-2 and, in RF, on average, 19.9–24.9% for L-1 and 13.7–16.3% for L-2.

Based on the preceding results, it is evident that, for predicting algal alert levels, the improvement in predictive accuracy via resampling methods surpassed that achieved by feature selection methods. Balanced predictions are made for each class. Despite the results of this study, feature selection methods can efficiently describe the input data while reducing the influence of noise or irrelevant variables, thereby providing better predictive results [55]. Moreover, in classification problems, using variables with a low statistical correlation to classes as pure noise can introduce bias in the prediction of classes and degrade the classification performance [56]. However, feature selection methods can be effective in improving the predictive performance of datasets with numerous features. In the present study, the number of features was 14, which is relatively small compared to the number of features used in previous studies. For example, Bolón-Canedo et al. [57] compared the predictive performance of various feature selection methods for 64 different datasets, with the number of features ranging from 918 to 41,151, and Wei et al. [58] studied 14 different datasets, with the number of features ranging from 72 to 400. Xue et al. [59] demonstrated that applying feature selection methods improved predictive performance with a reduced number of features. However, they also reported that the application of feature selection methods did not significantly enhance the predictive performance of models that already exhibited high accuracy. Therefore, in terms of data with imbalances and fewer features, the application of resampling methods may be more effective than feature selection methods in improving model predictive performance.

Appendix A present the results of the optimal models selected from the models that were iteratively performed for ANN and RF, respectively. The comparison of the performance of the optimal models exhibited results similar to the overall performance comparison. Based on these results, the best model was identified for predicting algal alert levels among the ANN and RF models. Figure 6 compares the confusion matrix between the selected optimal ANN model and the model using the original data. Non-linear feature selection and ENN sampling were applied to data from the selected ANN model. In the training step, accuracy was similar and the recall for L-1 increased from 63.6 to 86.2%, while the recall for L-2 increased from 81.0 to 90.5%. In the test step, despite a decrease in accuracy by 9.8%, the recall for L-1 increased from 64.3 to 71.4%, achieving balanced predictions at each algal alert level. Figure 7 shows the results of the confusion matrix comparison of the optimal RF model selected from the RF models with various data types.

The selected optimal RF model was constructed from data obtained using the ENN sampling method without feature selection. In the training step, the optimal model showed an increase of 11.3% in accuracy compared with the model using original data. During the test step, although the accuracy decreased by 1.9%, the recall for L-1 and L-2 increased by 35.7% and 50.0%, respectively. The recall of each algal alert level was as follows: 85.0% for L-0, 85.7% for L-1, and 100% for L-2. The hyperparameters of the optimal ANN and RF models are listed in Appendix A. In this study, the model with ENN sampling was selected as the optimal model for predicting algal alert levels. A comparison of the optimal models revealed that the models using non-linear feature selection and the CC sampling method exhibited balanced predictions compared with the models using the original data (Appendix A). All performance indices for both the training and test steps were higher for the optimal RF model than for the optimal ANN model (Figure 6 and Figure 7). Therefore, the RF model was deemed more suitable for predicting the algal alert levels.

## 4. Conclusions

This study presented a series of processes for improving the prediction of algal alert levels in the BJR. Based on the observed data, feature selection and resampling methods were applied and two machine learning models were constructed. The following major conclusions were drawn from this study:Applying resampling methods to the imbalanced classes observed in the original data allowed the collection of data with balanced distributions for all classes, thereby preventing biased learning of the model and improving its accuracy.Resolving the class imbalance via resampling methods proved to be more effective in improving the accuracy of the model than adjusting the input variables via feature selection methods.In the RF model, the accuracy of the model with the resampling method demonstrated the highest performance, whereas in the ANN model, the predictive performance of the model incorporating both feature selection and resampling methods appeared to be superior.When considering non-linear models such as machine learning for prediction, it is important to evaluate the availability of feature selection and resampling methods according to the model type.The characteristics and quantity of the original data can serve as important factors when selecting the feature selection and resampling methods. In addition, appropriate feature selection and resampling methods can be applied as useful tools for constructing machine learning models.

This study aimed to construct a prediction model for algal alert levels in reservoirs using readily available data from national monitoring stations and to provide a machine learning model that improves accuracy via feature selection and resampling methods. The proposed model is expected to be useful to engineers and decision makers involved in the management of algal blooms in watershed areas, including inland weirs, facilitating the establishment of effective strategies and regulations for their construction and operation.

## Figures and Tables

**Figure 1 toxics-11-00955-f001:**
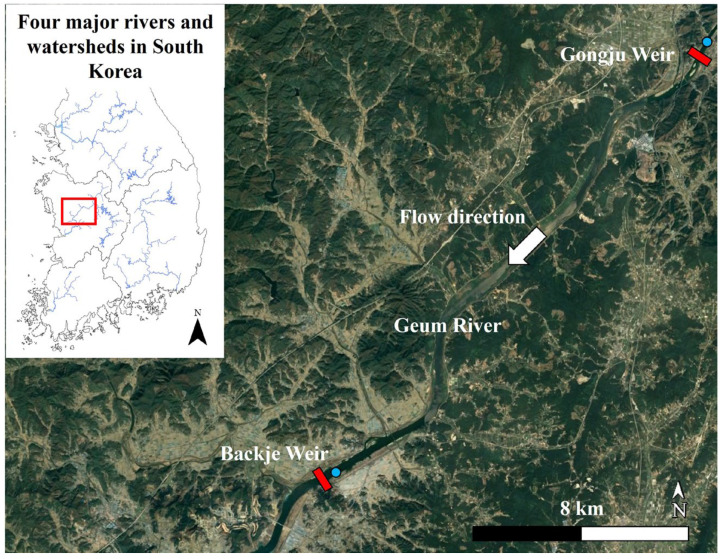
Map of the study area (126°56′20″ E–127°05′55″ E and 36°19′07″ N–36°27′45″ N) showing the Backje weir dam, Gongju weir dam, and water quality monitoring stations. The red box is a weir dam installed in Geum River and the blue circles are water quality monitoring stations that measure the algal cell density and water quality variables.

**Figure 2 toxics-11-00955-f002:**
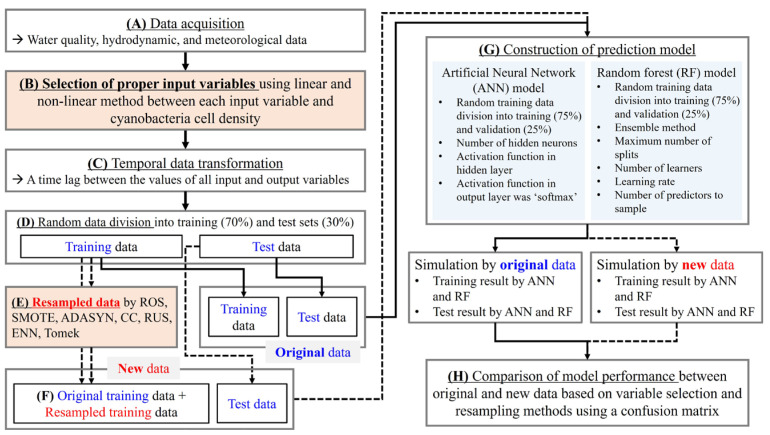
Flow chart for the construction of artificial neural network (ANN) and random forest (RF) models to predict algal alert levels using original and synthetic data according to variable selection and resampling methods.

**Figure 3 toxics-11-00955-f003:**
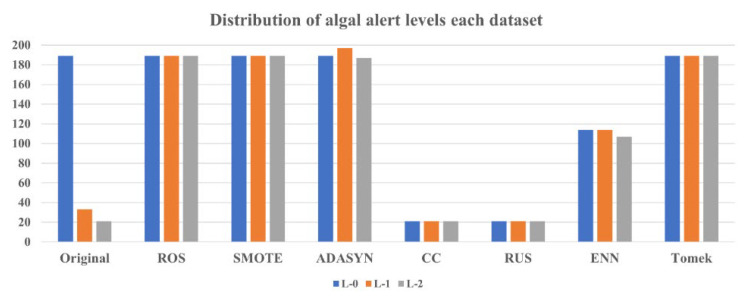
Comparison of algal alert level distribution between the original data and the new dataset with resampled data via each sampling method.

**Figure 4 toxics-11-00955-f004:**
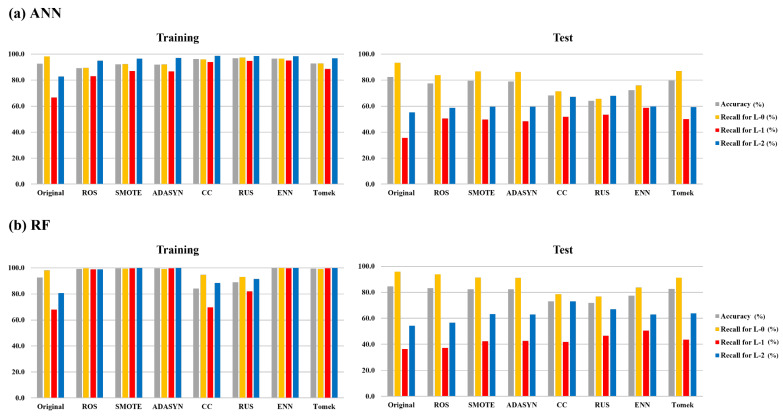
Comparison of overall prediction performance for algal alert levels of applied resampling methods based on original dataset. (**a**) is the overall prediction performance in ANN and (**b**) is the overall prediction performance in RF. Original is a model in which the original data were used, ROS is a model with random oversampling method, SMOTE is a model with synthetic minority oversampling technique, ADASYN is a model with adaptive synthetic sampling, CC is a model with cluster centroid undersampling, RUS is a model with random undersampling, ENN is a model with synthetic minority oversampling technique–edited nearest neighbor, and Tomek is a model with synthetic minority oversampling technique–Tomek link.

**Figure 5 toxics-11-00955-f005:**
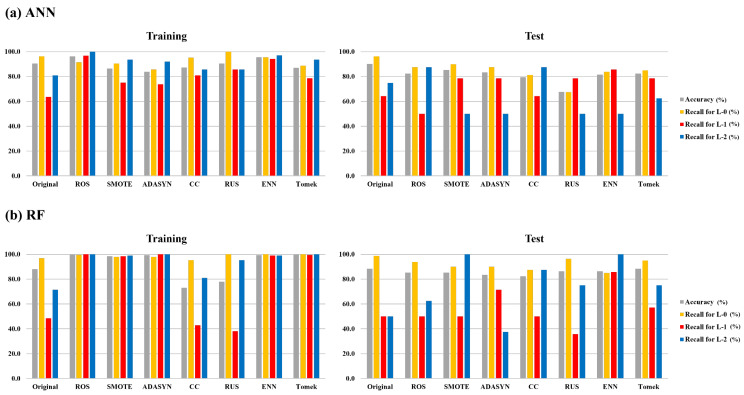
Comparison of prediction performance for algal alert levels of the optimal model by applied resampling methods based on the original dataset. (**a**) is prediction performance in ANN optimal model and (**b**) is prediction performance in RF optimal model. Original is a model in which the original data were used, ROS is a model with random oversampling method, SMOTE is a model with synthetic minority oversampling technique, ADASYN is a model with adaptive synthetic sampling, CC is a model with cluster centroid undersampling, RUS is a model with random undersampling, ENN is a model with synthetic minority oversampling technique–edited nearest neighbor, and Tomek is a model with synthetic minority oversampling technique–Tomek link.

**Figure 6 toxics-11-00955-f006:**
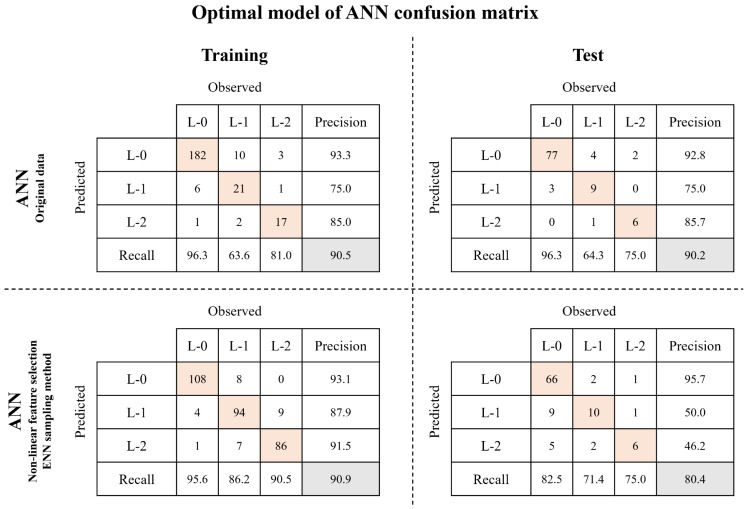
Comparison of confusion matrices between ANN model using original data and selected optimal ANN model.

**Figure 7 toxics-11-00955-f007:**
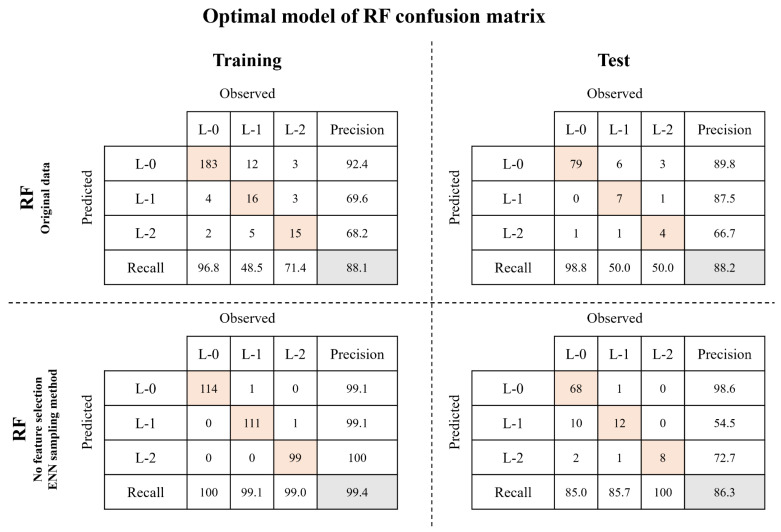
Comparison of confusion matrices between RF model using original data and selected optimal RF model.

**Table 1 toxics-11-00955-t001:** Statistical analysis and variable selection results of 14 input variables collected for the prediction of alert level from 2013 to 2021 at the BJR.

Variables		Description	Unit	Descriptive Analysis		Variable Selection Method	
				Range	Mean	Dependence Test (*p*-Value)	MI Score
Water quality	TDN	Total dissolved nitrogen concentration	mg/L	1.17 to 6.92	2.79	<0.001	0.128
	NO_3_-N	Nitrate concentration	mg/L	0.72 to 3.91	2.13	<0.001	0.118
	NH_4_-N	Ammonium concentration	mg/L	0.01 to 2.24	0.19	0.005	0.015
	TDP	Total dissolved phosphorus concentration	mg/L	0.01 to 0.16	0.04	0.001	0.085
	PO_4_-P	Phosphate concentration	mg/L	0 to 0.15	0.02	<0.001	0
	Cond	Conductivity	μmhos/cm	125 to 639	348.23	0.001	0.012
	GJ-cell	Cyanobacteria cell density in GJR	cells/mL	0 to 50970	1077	<0.001	0.161
Hydro-dynamic	Wlevel	Average water level of the BJR	m	1.21 to 5.01	3.71	0.396	0.026
	Inflow	Average inflow rate of the BJR	m^3^/s	20.25 to 2536.23	145.59	0.011	0.044
	Discharge	Average total discharge rate of the BJR	m^3^/s	20.10 to 2555.66	145.70	0.011	0.049
	Dhydro	Average discharge rate by the hydropower plant of the BJR	m^3^/s	0 to 124.60	43.85	0.050	0.043
Meteorological	Atemp	Average air temperature	°C	−1.30 to 30.46	16.85	<0.001	0.207
	Precip	Accumulated precipitation	mm	0 to 352.80	28.38	0.934	0.016
	Wspeed	Average wind speed	m/s	0.56 to 2.39	1.30	0.587	0

**Table 2 toxics-11-00955-t002:** Results of descriptive analysis for monthly cyanobacteria cell density and nutrient concentration measured in the BJR.

Month	Algal Alert Level (Number of Events)			Cyanobacteria Cell (Cells/mL)		N:P Ratio		Chl-a (µg/L)		Phosphate (µg/L)	
	L-0	L-1	L-2	Range	Average	Range	Average	Range	Average	Range	Average
March	40	0	0	0 to 140	4	29.1 to 240.8	74.8	7.6 to 105.3	42.3	1 to 29	6.7
April	39	0	0	0 to 625	34	16.4 to 139.2	55.3	16.3 to 162.8	56.6	0 to 43	7.5
May	36	2	0	0 to 1950	117	8.7 to 123.4	39.9	20.2 to 176.1	64.5	1 to 113	10.9
June	38	1	0	0 to 2920	131	12.2 to 50.6	30.0	11.9 to 185.1	70.8	0 to 33	9.0
July	25	5	7	0 to 95,500	7684	6.5 to 46.8	23.8	7.4 to 165.1	46.2	2 to 140	32.2
August	12	10	17	0 to 398,820	27,391	5.3 to 42.9	19.6	5.3 to 144.3	51.1	2 to 135	40.4
September	16	15	5	0 to 95,355	7206	6.8 to 84.5	27.9	6.4 to 177.7	55.1	2 to 153	34.1
October	25	13	0	0 to 6565	1071	7.4 to 84.1	43.6	9.3 to 123.0	49.8	1 to 141	20.8
November	38	1	0	0 to 1160	51	14.9 to 135.8	65.2	5.1 to 128.4	35.7	1 to 97	15.5
Total	269	47	29	0 to 398,820	4828	5.26 to 240.8	42.5	5.1 to 185.1	52.4	0 to 153	19.5

**Table 3 toxics-11-00955-t003:** Comparison of overall and optimal performance for accuracy, recall, and precision for algal alert levels according to the feature selection methods in ANN and RF.

Machine Learning Model		Feature Selection	Training (Including Validation)							Test						
			Performance Index							Performance Index						
			Accuracy	Recall			Precision			Accuracy	Recall			Precision		
				L-0	L-1	L-2	L-0	L-1	L-2		L-0	L-1	L-2	L-0	L-1	L-2
Overall model performance	ANN	No feature selection	92.6 (±6)	98.3 (±2)	66.6 (±30)	82.8 (±17)	94.1 (±5)	82.5 (±17)	87.6 (±12)	82.4 (±3)	93.3 (±4)	35.6 (±12)	55.3 (±17)	88.9 (±2)	55.6 (±16)	56.3 (±16)
		Linear method	91.7 (±6)	98.1 (±1)	62.7 (±27)	79.3 (±16)	93.4 (±5)	80.8 (±16)	85.0 (±12)	83.0 (±3)	93.7 (±4)	38.2 (±13)	54.9 (±16)	89.2 (±3)	58.0 (±17)	60.0 (±16)
		Non-linear method	91.2 (±6)	98.2 (±1)	59.8 (±29)	78.5 (±17)	93.1 (±5)	78.8 (±16)	84.2 (±11)	82.2 (±4)	93.4 (±5)	34.9 (±13)	53.3 (±19)	89.0 (±2)	54.0 (±18)	56.7 (±16)
	RF	No feature selection	92.5 (±4)	98.1 (±1)	67.9 (±18)	80.6 (±15)	94.2 (±3)	83.1 (±13)	88.5 (±10)	84.4 (±2)	95.8 (±2)	36.4 (±11)	54.4 (±16)	89.7 (±2)	53.8 (±15)	67.5 (±17)
		Linear method	92.4 (±4)	98.1 (±1)	67.5 (±17)	80.5 (±14)	94.2 (±3)	82.8 (±12)	87.2 (±11)	85.2 (±2)	96.0 (±2)	40.1 (±11)	56.9 (±16)	90.2 (±2)	60.2 (±15)	66.8 (±15)
		Non-linear method	93.5 (±4)	98.3 (±2)	72.1 (±18)	83.1 (±14)	95.0 (±3)	85.5 (±13)	89.5 (±11)	84.5 (±3)	95.3 (±3)	38.3 (±13)	56.8 (±17)	89.9 (±2)	55.6 (±14)	65.4 (±15)
Optimal model performance	ANN	No feature selection	90.5	96.3	63.6	81.0	93.3	75.0	85.0	90.2	96.3	64.3	75.0	92.8	75.0	85.7
		Linear method	84.0	96.8	27.3	57.1	88.0	50.0	70.6	88.2	98.8	42.9	62.5	89.8	85.7	71.4
		Non-linear method	84.8	96.3	24.2	76.2	87.5	53.3	80.0	88.2	98.8	50.0	50.0	88.8	87.5	80.0
	RF	No feature selection	88.1	96.8	48.5	71.4	92.4	69.6	68.2	88.2	98.8	50.0	50.0	89.8	87.5	66.7
		Linear method	94.7	98.9	75.8	85.7	94.9	89.3	100	89.2	98.8	50.0	62.5	89.8	100	71.4
		Non-linear method	94.7	97.9	81.8	85.7	95.9	87.1	94.7	92.2	100	42.9	100	92.0	100	88.9

## Data Availability

Water quality data can be downloaded from the Water Environment Information System (https://water.nier.go.kr/web; accessed on 22 November 2023). Meteorological data can be downloaded from the Korea Meteorological Administration (https://www.weather.go.kr/w/index.do; accessed on 22 November 2023). Hydrological data can be downloaded from the Korea Water Resource Corporation (https://www.water.or.kr/kor/realtime/sumun/index.do?mode=mult&menuId=13_91_93_98; accessed on 22 November 2023).

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
