# Peer review of "Machine Learning-Based Early Warning Level Prediction for Cyanobacterial Blooms Using Environmental Variable Selection and Data Resampling"

_toxics, 2023, doi:10.3390/toxics11120955_

Round 1
Reviewer 1 Report
Comments and Suggestions for Authors
The MS explored machine learning-based early warning level prediction for cyanobacterial blooms using environmental variable selection, the topic is very meaningful, but also difficult. The following concerns should be addressed before it goes to any future.
(1) For section ABSTRACT, there are so many backgrounds, the section ABSTRACT is mainly divided into four parts: purpose, method, results and conclusions. I suggest to simplify the ABSTRACT in current form.
(2) Lines 61 -63 on page 1, “The recent rise in water temperature owing to climate change and the increase in nutrient discharge caused by human activity have promoted the growth of HABs in aquatic ecosystems.” The following references could be cited to support the statements: https://doi.org/10.1016/j.ocecoaman.2023.106554.
(3) I understand the work done by the author, using machine learning-based to establish the relationship between cyanobacteria concentration and some factors, but how to determine whether it is a disaster?
(4) For hydro-dynamic in Table 1, why didn’t the author consider the water temperature? You know, this may be one of the important factors that cause HABs.
(5) There are many machine learning models, why the author chose ANN model and RF model? Corresponding reasons should be given.
Reviewer 2 Report
Comments and Suggestions for Authors
This is a very valuable research and has certain application prospects. The prediction of algal blooms, especially toxic algal blooms, is a global challenge, and there is still no good method or technology to date. The author has conducted a good exploration, and if further modifications and additional information can be made, I believe it would be more worthwhile to be published.
1) “A total of 429 cyanobacterial cell density and water quality data”, what specific data are included separately and the quantity of each data?
2) Are these data sufficient for machine learning and how are they evaluated?
3) Please unify and further check the format of the units in Table 1 and the text, as there are many errors.
4) I suggest the author add a few cases for predictive validation, to demonstrate the accuracy and reliability of the method.
Comments on the Quality of English Languageno
Reviewer 3 Report
Comments and Suggestions for Authors
The work presented uses ML techniques to improve the preduction of cyanobacterial blooms. The paper is well written and the work is well designed. I will make only a couple of comments:
- Some paragraphs in the “Results” section should be better located on other parts of the paper. For instance in section 3.5 from lines 579-590 must be move to the introduction (from line 150).
- I would also move lines 591-599 a bit before in section 3.5, like on line 549.
- Point out before in the text that the ENN sampling was optimal. This sentence appears to late in the result section.
- Add figure S2 in section 3.3, I think it will help to clarify what is written.
- Since some results contradict previous published works, authors should make an effort to explain which other variables could bi influencing their own results: for instance if winds speed is not a relevant feature, maybe some other physical variables like the shape of the water mass or even the wind direction, are having an effect.
Reviewer 4 Report
Comments and Suggestions for Authors
Review of the manuscript titled “Machine learning-based early warning level prediction for cyanobacterial blooms using environmental variable selection and data resampling”.
The article covers a topic on which many publications have already been published, but due to the growing threat of cyanobacterial blooms, it requires further research and in-depth analyses. In particular, research into predicting and preventing toxic algae blooms is important.
The results obtained during the analyzes described in the manuscript are innovative and worth publishing.
The model was tested on data from the dam reservoir. Information about this should be included in the title and abstract. Results may vary in different types of water.
D
etailed comments:
Introduction
I suggest shortening the first paragraph. Details about health effects in individual cases seem unnecessary.
Materials and methods
Section 2.1.
Fig. 1 Small map on the left side not clear. No world directions marked.
Section 2.2.
Lines 185-186 It was 429 samples collected or different sampling points? It is not clear.
The WHO risk thresholds of cyanobacterial blooms are popular in the scientific literature https://iris.who.int/bitstream/handle/10665/42591/9241545801.pdf?sequence=1 (Guidelines for safe recreational water environments VOLUME 1 COASTAL AND FRESH WATERS, 2003).
Is it possible to compare these threshold values with the values (caution and warning) used in the manuscript?
Section 2.3
No numerical references to literature (Croxton and Cowden, 1939; Ross, 2014). This situation also occurs elsewhere in the text.
Results
Fig. 3
Please add an explanation of the abbreviations or information where they are located. The graph should be self-explanatory.
Round 2
Reviewer 1 Report
Comments and Suggestions for Authors
The concerns have been addressed, and the MS is acceptable in current form.
Reviewer 2 Report
Comments and Suggestions for Authors
Thank you for the author's reply.
I have no further comments. I suggest publishing.